# Spectral Acoustic Fingerprints of Sand and Sandstone Sea Bottoms

Uri Kushnir * and Vladimir Frid *

Civil Engineering Department, Sami Shamoon College of Engineering, Ashdod 77245, Israel
* Correspondence: uriku@ac.sce.ac.il (U.K.); vladimirf@ac.sce.ac.il (V.F.)

**Abstract:** Modern studies which dealt with the frequency domain analysis showed that a frequency-domain approach has an essential advantage and mentioned an inner qualitative relationship between the subsurface structure and its frequency spectra. This paper deals with the acoustic spectral response of sand and sandstone sediments at the sea bottom. An acoustic data collection campaign was conducted over two sand sites and two sandstone sites. The analysis of the results shows that reflections of acoustic signals from sand and sandstone sea bottom are characterized by various spectral features in the 2.75–6.75 kHz range. The differences in acoustic response of sand and sandstone can be quantified by examining the maximal normalized reflected power, the mean frequency, and the number of crossings at different power levels. The statistical value distribution of these potential classifiers was calculated and analyzed. These classifiers, and especially the roughness of the spectrum quantified by the number of crossings parameter can give information to assess the probability for sand or sandstone based on the reflected spectra and be used for actual distinction between sand and sandstone in sub bottom profiler data collection campaigns.

**Keywords:** marine survey; acoustic reflection; spectral analysis; sediments identification

## 1. Introduction

Acoustic waves are the method of choice for remote sensing in the aquatic environment. This is true for a soil investigation including bathymetric mapping (topography of the ocean floor), shallow and deep acoustic surveys of the subsoil, identification of buried and on-ground existing infrastructure, and underwater positioning [1–12]. The industry-standard remote sensing tools available to gain insight on the sea-bottom upper layer composition relay on calibrated backscatter intensity. Analysis of the scientific literature demonstrates that the amount of more advanced methods of research conducted regarding the spectral analysis of the acoustic response of the sea bottom is limited and refers mostly to large swath, non-localized side scan data and rarely to single beam type sonars including sub bottom profilers or multibeam sonars. Signal processing as well as image processing techniques were used, sometimes with combined AI algorithms. However, these do not yet offer an applicable quantitative method for classification [13–28]. Other studies focused on calibrated backscatter intensity methods [29,30], sometimes assisted with a sub-bottom-profiler for geological background [31]. Our analysis portrays that most worldwide acoustic studies of the seabed aimed at assessing the sub-bottom soil composition are based on backscatter intensity, e.g., [32], and extensive, area-specific calibration i.e., the geocoder algorithm [33]. Anderson et al. [34] in their comprehensive review clearly stated that without careful calibration it is doubtful that statistically based interpretations relating acoustic-backscattering measurements to the seabed's sub-surface properties and content can be achieved. The few studies that have been conducted to identify the type of sea-bottom soils based on the spectral features were utilized mostly on side-scan sonar data. The problem is that due to its method of operation, the spectral features of side scan sonar data reflect large-scale relief features [27]. In addition, a single swath may include several

types of soil. Hence, such methods are not general enough and although they may provide good insight over large homogenous sea bottom areas, they are less efficient in areas which are comprised of alternating types of soil (rock and sand, for example).

Demarco et al. [35] presented an evaluation of the response of seismic reflection attributes in diverse types of marine substrate (rock, shallow gas, sediments) using seafloor samples for ground-truth statistical comparisons. The spectral-ratio technique presented in [36] for seismic-reflection data shows that a frequency-independent quality factor Q is the most proper fit to unconsolidated sand- and clay-dominated sediments in the frequency range of 0.5–8.0 kHz. LeBlanc et al. [37] used Chirp sub bottom profiler calibrated digitally recorded data to estimate surficial acoustic reflection coefficients as well as a complete sediment acoustic impedance and to predict sediment properties.

From the analysis of the current state-of-the-art, it can be seen that the approaches related to the topic of the article are based on the use of a high-frequency echo sounder, which differs from the system used in the presented study. Using low frequency (CHIRP based) methodologies are rare (e.g., [35,36]) In addition, a statistical analysis of a large data set that can identify classifiers that are statistically separate for diverse types of soil had yet to be found. As can be seen from the state-of-the-art examination presented above, the knowledge gap consists of three main lacunas: (a) The lack of spectral and statistical analysis of large data sets collected either by single beam chirp sonars or the separate beams of chirp multibeam sonars (both are readily available), which will reflect the intrinsic properties of the soil rather than large scale relief/bathymetric features (as the case with side-scan sonar data); (b) The lack of identification of statistically separate characteristics of the reflected spectrum from different soil types, which could be used as classifiers; and (c) The lack of understanding of the specific mechanisms of acoustic wave propagation and reflection within actual soil structures, which results in the differences in classifier values for diverse types of soil.

Due to these deficiencies, a validated method for determining marine soil based on the analysis of spectral features of acoustic signals is not yet available.

The main work hypothesis of the research consisted in understanding that spectral features of acoustic signals reflected from the sand and sandstone sea bottoms are associated with their physical peculiarities. These singularities include fine-scale topography at the top of both types of sediments as well as the heterogeneity of several meters (depending on the signal length) below the top of the reflector. Being significantly dissimilar for sand and sandstone these singularities are expected to affect the acoustic signals reflected from the top of the sea bottom and hence induce dissimilarities in the spectral parameters (e.g., amplitude, main frequency, the frequency-dependent reflection coefficient, number of spikes, etc.). The objectives of the paper are to study the spectral features of acoustic signals reflected from the sand and sandstone at the sea bottom and to identify, quantify, and examine the efficiency of spectral attributes that can be used for practical purposes of classification. This study is a first, but an essential step of critical importance in characterizing sand and sandstone reflectors where other imaging technologies (e.g., side scan and multi-beam sonars) are not available.

## 2. Method

The method adopted for this paper includes a preliminary controlled collection of the sea bottom reflections of frequency-modulated (hereby: FM) acoustic signals transmitted by a chirp sub-bottom-profiler (hereby: SBP).

The data was collected over two verified sandstone sites and over two verified sandy bottom sites and recorded in SegY format. The details on the sites under study and instrument used for the data acquisition are presented below in Section 3. The procedures of data analysis and quality control are presented below in the corresponding sections. In particular, the reflected signals were transformed to the frequency domain by the FFT procedure (Section 4.3). Based on the resulted spectra, several potential classifiers were identified including maximal normalized power, mean frequency, and the spectrum

roughness quantified by the number of crossings at certain power levels. The statistical study of the potential classifiers was conducted in two approaches. First, the probability to receive a certain classifier value was calculated and examined. Second, for a given value of a classifier, the probability that the reflector is either sand or sandstone was calculated. These results were the basis of quantifying the efficiency of the classifiers in distinguishing between a sandy sea bottom and a sandstone sea bottom.

## 3. Experimental Setup

### 3.1. Acoustic Device of Choice

The acoustic device chosen for the data collection campaign was the Bathy-2010PC chirp sub-bottom profiler (SBP). This device is a linear chirp with a linear frequency swipe from 2.75 kHz to 6.75 kHz with a signal duration of 5.4 ms and a beam angle of 30°. In this setting, the fine-scale topography of the reflector (boulders, sand ripples, etc.) is expected to influence the reflected signal.

### 3.2. Data Collection Sites

The data was collected offshore of Ashkelon from two sites (hereby "site 1 and site 2") approximately 1200 m apart, with a known sandy bottom at a depth 26 m and over two sites (hereby "site 3 and site 4") approximately 800 m apart, with known sandstone bottom at a depth of 33 m (soil composition at the sites was verified in a previous site survey by means of sand sampling and a drop camera (See Figure 1). Grain size distribution analysis showed that the sand could be categorized as poorly graded sand, (SP), based on the Unified Soil Classification System (USCS). The range of grain sizes is 0.075 mm–2.36 mm, while 95% of the grain size belongs to the interval 0.15–0.6 mm. The sandstone properties are presented in [38]. The depth was kept similar in the site pairs to minimize the effect of depth-related attenuation. All sites include an area of at least a 200 m radius of similar soil composition.

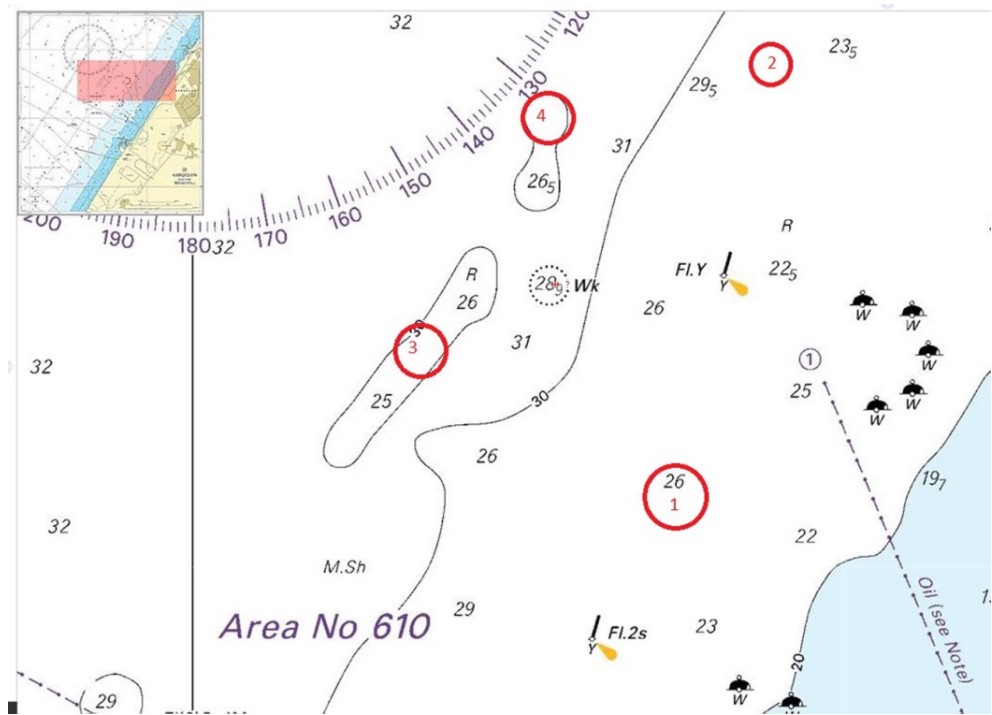

**Figure 1.** Data collection sites: (1,2)—Sand sites at 26 m. (3,4)—Sandstone sites at 33 m.

### 3.3. Vessel Setup and Transmission Parameters

During data collection at each site, the vessel was moored and shifted position slightly with the slightly changing wind and wave direction. Hence, each trace was shot at a slightly different location, however over the same soil type. The transducer was installed over

the side at a depth of 1m below the waterline. The sound speed measured in water was 1530 m/s. The data was recorded raw (i.e., without chirp compression). For each trace, a total of 100 ms were recorded. This includes the transmitted signal, water column, the bottom, and about 120 m of soil penetration. The transmission parameters are summarized in Table 1.

**Table 1.** Transmission Parameters.

| Site | Site 1 | Site 2 | Site 3 | Site 4 |
|---|---|---|---|---|
| Soil Type | Sand | Sand | Sandstone | Sandstone |
| Depth | 26 m | 26 m | 33 m | 33 m |
| Transducer depth | 1 m | 1 m | 1 m | 1 m |
| Transmission power | −18 dB | −18 dB | −18 dB | −18 dB |
| Water Sound Velocity (m/s) | 1530 | 1530 | 1530 | 1530 |
| Recorded Signal duration (ms) | 100 | 100 | 100 | 100 |

## 4. Data Analysis

The data from the SegY files was extracted using the SegyMat subroutines in MATLAB for further analysis. This included a repeatability check for the transmitted signal, signal separation (transmitted signal, Reverberations, and first reflection), spectral analysis of the first reflection, and then the statistical analysis of spectral features.

### 4.1. Quality Control

To validate that the differences in the reflected signal are due to soil characteristics and not from changes in the transmitted signal—the signal of all the traces was checked for repeatability. This was done for all traces and all 4 sites, and the results demonstrate almost absolute repeatability of the transmitted signal. The signal-to-noise (S/N) ratio varied between 15–40. The value of reflection coefficient for four types of sites cannot be properly defined since the most transmitted signal was saturated. However, the value of the first half cycle of both signals (transmitted and reflected) was not saturated. That is why the average value of the reflected coefficient for both types of sites was calculated as the ratio of amplitude of the first half cycle of reflected to transmitted signals (including water column attenuation): 0.05 and 0.15, respectively, for sand and sandstone sites.

### 4.2. Transmission, Reverberations, and First Reflection Separation

The next stage was the separation of the time series to the transmitted part, the reverberation's part, the first reflection, and the rest of the signal. In this research, we focus on the first reflection in order to characterize the first layer of soil. The transmitted signal and the first reflection were extracted from each trace. The transmitted signal includes the first 5.4 ms of each trace. It is noted that the transmitted signal, which would preferably be used to normalize the reflected signal, as could be expected in these circumstances. Hence, it cannot be used for normalization. The first reflection extraction is needed to identify the start point of the reflection. This was done by finding the sharp increase of the rate of growth of the size of the local extremum values characterizing the start of the reflection, as appears in Figure 2. The duration of the first reflection is equal to that of the transmitted signal. Figure 3 shows the enlarged signatures of the first arrival of the stacked reflected signal at the four sites.

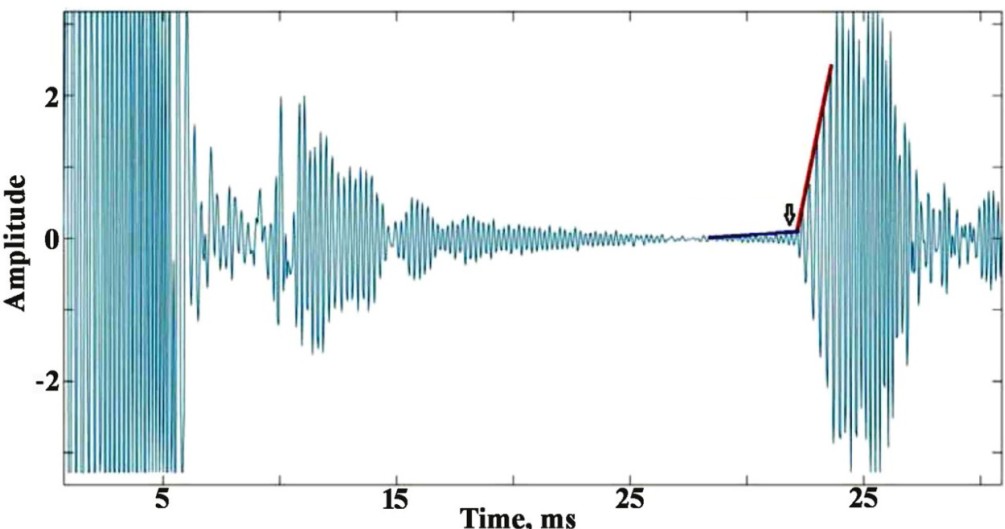

**Figure 2.** Identification of the beginning of the first reflection. The arrow in the picture shows the first arrival of the reflected signal.

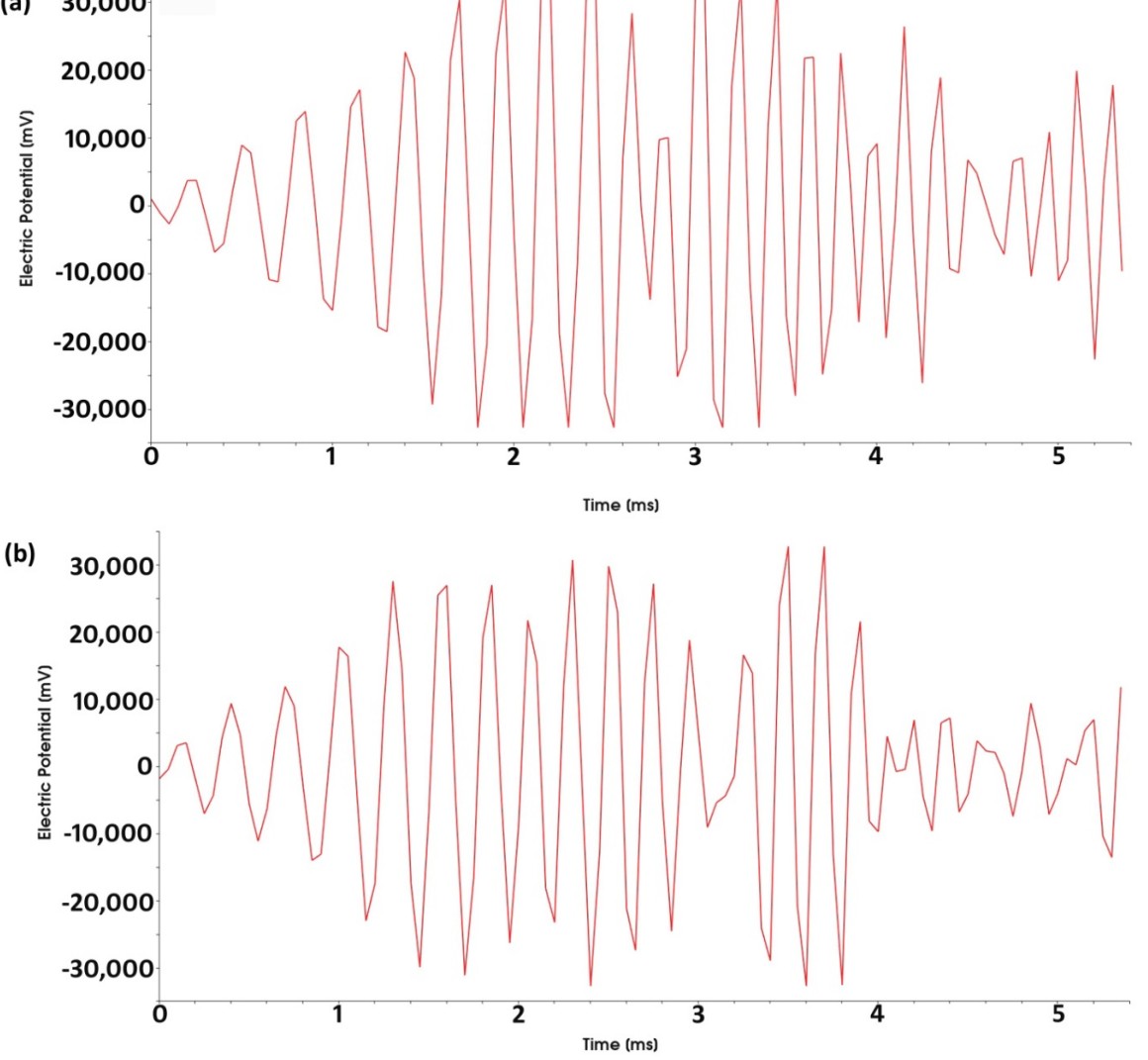

**Figure 3.** *Cont.*

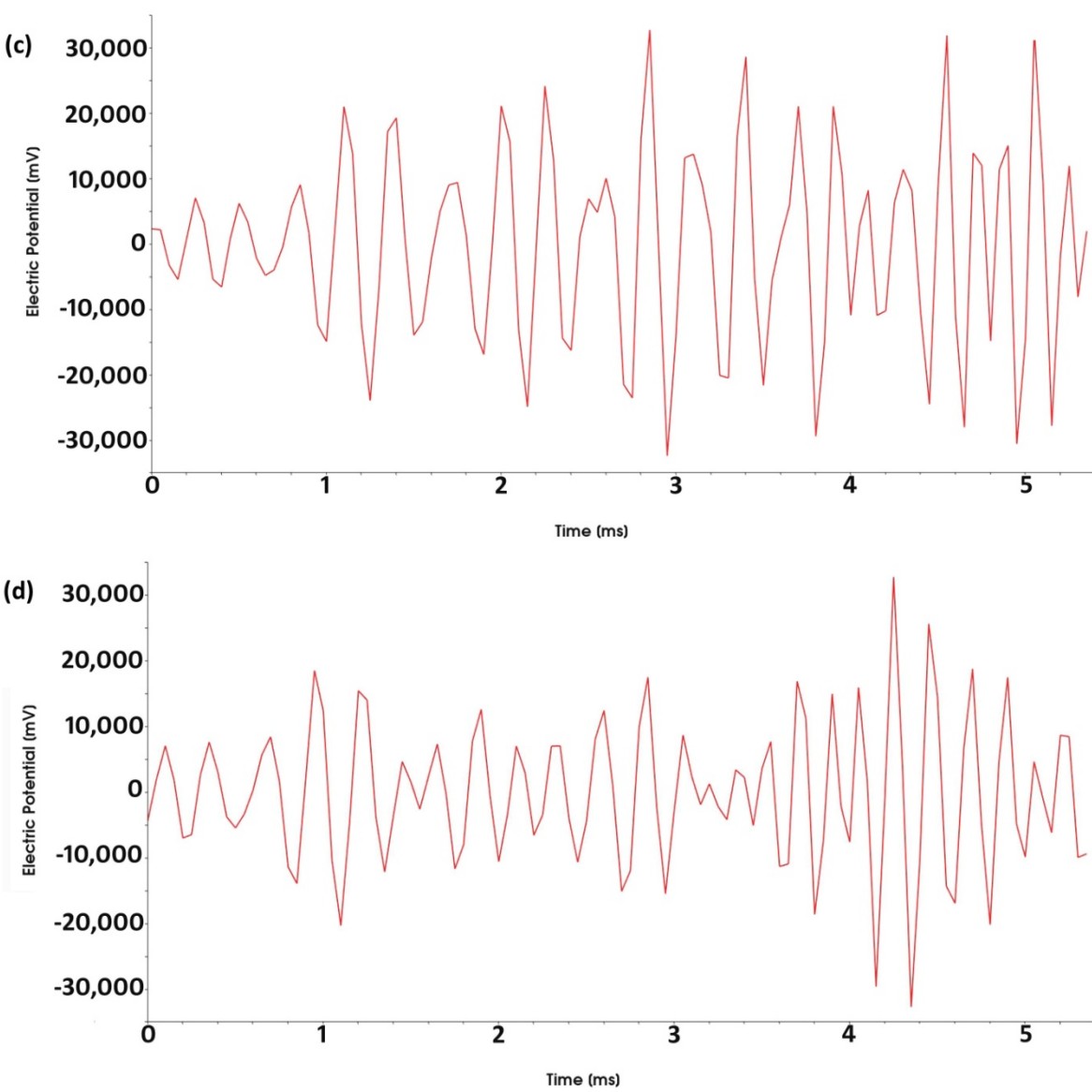

**Figure 3.** Enlarged signatures of first arrival of the reflected signal. (**a**) Site 1—sand. (**b**) Site 2—sand. (**c**) Site 3—Sandstone. (**d**) Site 4—Sandstone.

### 4.3. Classifier Statistical Analysis

Once the first reflection time series for every trace was extracted, the autocorrelation sequence for each trace was calculated. The Discrete Fourier Transform of the autocorrelation was computed to obtain the coefficients in the frequency domain according to the following equations:

$$F(n) = \sum_{i=0}^{N-1} x_i e^{-\frac{2\pi j}{N} i n} \tag{1}$$

where is $x_i$ the sequence of length N. The MATLAB uses the FFTW library to perform the Fourier function (http://fftw.org/, accessed on 2 November 2022). In FFTW, the calculation of the transformed data is performed by an executor, consisting of blocks of C code called "codelets". Each codelet specializes in one part of the transformation. With these codelets, the performer implements the Cooley-Turkey FFT algorithm, which considers the size of the input signal. In recursive factoring, the signal is split into shorter parts. The results of transformations of short parts are multiplied, and, finally, the transformation of the original signal is calculated. The well-known algorithm is described in detail in numerous scientific papers and hereafter we cite only a few examples [39–46]. The one-sided power spectrum

was calculated by taking the positive frequency range only and multiplying the power values by two (except the first term) and then normalized by its area. The normalization was performed to cancel out the differences in attenuation between sand sites and the sandstone sites.

To gain first insight on the differences between reflections from sand and from sandstone and enhance the S/N ratio stacking of the reflection segment of the traces at each site was performed. Stacking is known to be a filtering procedure of "averaging" a set of repeated signals to diminish accidental noise for improved interpretation. Based on the shape differences between the stacked reflected spectra of the sand vs. the sandstone, potential classifiers were found and computed for each individual trace. Then, the classifiers were statistically analyzed to examine if there exists a separation in the probability distribution of the potential classifiers between the reflections from sand and from sandstone.

## 5. Results

Figure 4a,b show the spectrums of the reflected stacked signals normalized (by the spectrum area). It is seen that spectra of signals reflected from the sand bottom are significantly different from those reflected from the sandstone bottom. The following essential differences can be noted: (a) individual peaks for the sand spectrum are significantly wider than for sandstone; (b) the difference in peak heights (within a specific spectrum) for sand is much less prominent than that observed for sandstone; (c) the normalized height of the main peak in the spectrum for sandstone is significantly higher than that of sand; (d) the spectra reflected from sandstone consists of a larger number of spikes while the spectra reflected from sand are smoother; (e) the maximal peak height in the sand spectrum are quite similar for the two sand sites while quite different (a factor of about 1.7) for the two sandstone sites.

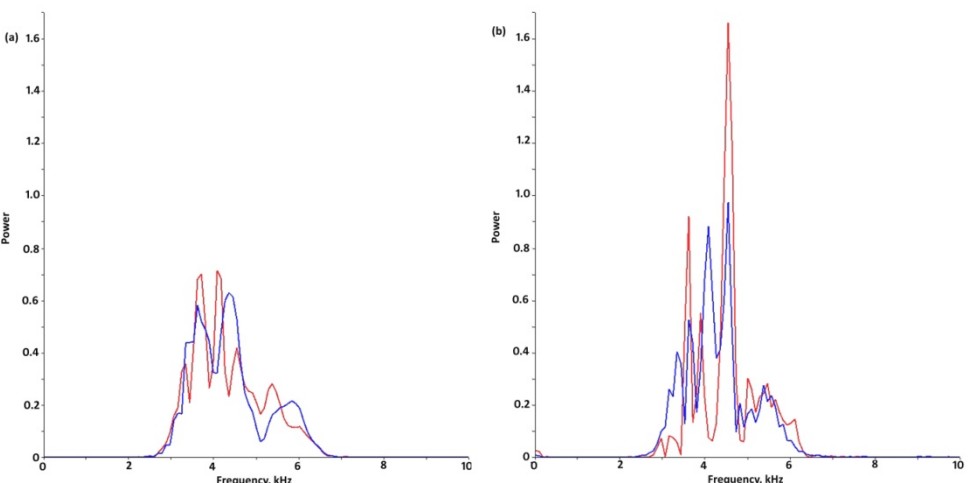

**Figure 4.** Stacked normalized power spectra [1/kHz]: (**a**) sand (site 1-red line), site 2 (blue line). (**b**) sandstone (site 3—red line), site 4 (blue line).

To quantify the above-noted differences we consider the following characteristics of the spectrum: (a) the probability distribution of the maximal normalized power; (b) the probability distribution of the mean frequency; (c) the probability distribution of the spectrum inertia about the frequency axis; (d) the probability distribution of the spectrum inertia about the power axis; (e) the probability distribution of the number of crossings at 1/12 of the maximal normalized power; and (f) the probability distribution of the number of crossings at 1/16 of the maximal normalized power (Figures 5 and 6). The following procedure was applied for the parameter's quantification: (a) for each parameter distribution and for each site the maximal value, the mode, the mean, and the width at the level of maximal value divided by factor "e" were calculated; (b) An equivalent area from the multiplication of the maximal value and the width was calculated; (c) the

variation between the distributions of the parameters for the different sites was estimated by integration of the root square difference (RSD), i.e.,:

$$\int \sqrt{\left(D_K^i(x) - D_K^J(x)\right)^2}\,dx \tag{2}$$

where $D_k^i$ denotes the distribution of the kth parameter for the ith site, with the parameter list being the maximal normalized power distribution, the mean frequency distribution, the spectrum inertia with respect to the x-x axis distribution, the spectrum inertia with respect to the y-y axis distribution, the number of crossings at the level of the maximal power divided by 12 distribution, and the number of crossings at the level of the maximal power divided by 16 distribution. The results are summarized in Tables 2–7.

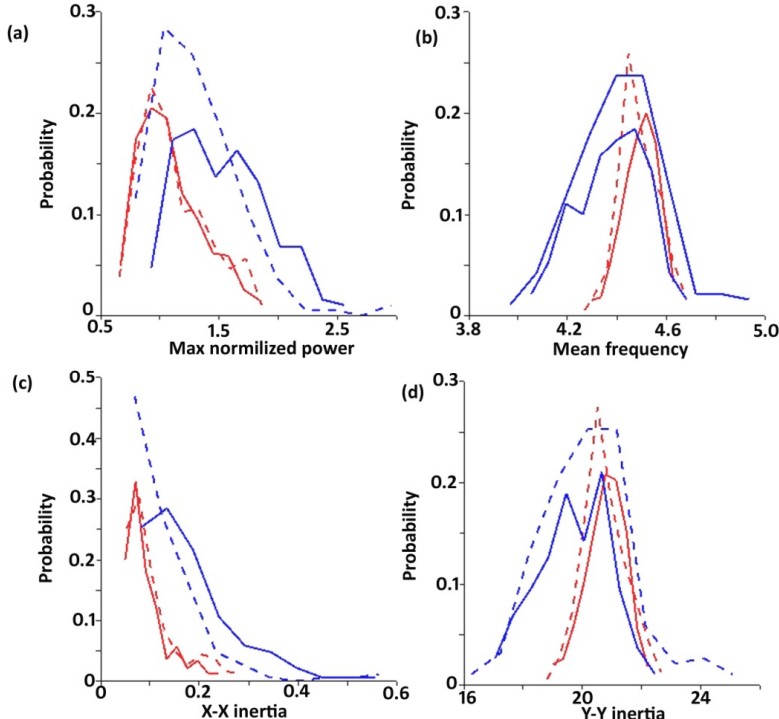

**Figure 5.** Probability distribution: (**a**) Maximal normalized power [1/kHz]. (**b**) Mean frequency; (**c**) Spectrum inertia about the x-axis. (**d**) Spectrum inertia about y-axis. Red full and dashed lines are sand site 1 and site 2, respectively, while blue full and dashed lines are sandstone site 3 and site 4, respectively.

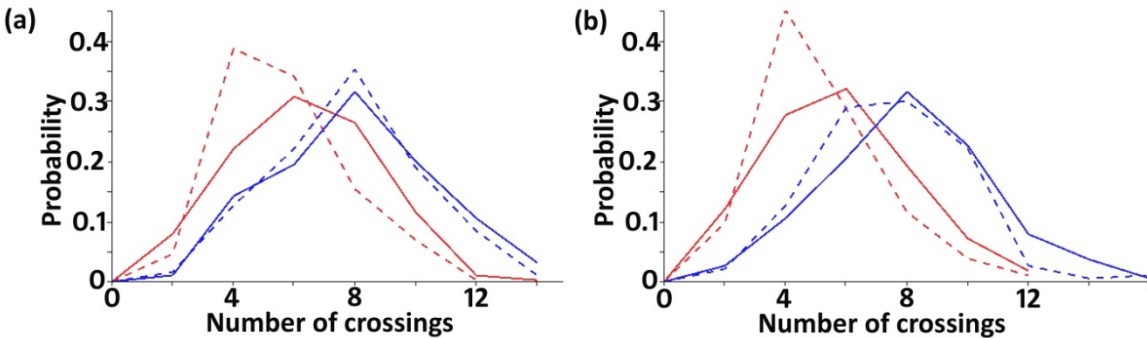

**Figure 6.** Probability distribution: (**a**) Number of crossings at 1/12 of the maximal normalized power. (**b**) Number of crossings at 1/16 of the maximal normalized power. Red full and dashed lines are sand site1 and site 2, respectively, while blue full and dashed lines are sandstone site 3 and site 4, respectively.

**Table 2.** Quality of the maximal normalized power as a classifier.

|  | Site 1 | Site 2 | Site 3 | Site 4 |
|---|---|---|---|---|
| Max | 0.20513 | 0.22564 | 0.18421 | 0.28421 |
| Width @ Max/e | 0.6976 | 0.6956 | 1.2370 | 0.9439 |
| MEAN | 1.0863 | 1.1082 | 1.5339 | 1.2932 |
| MODE | 0.9231 | 0.9213 | 1.2854 | 1.0295 |
| RSD WITH SITE 1 | NA | 0.0126 | 0.1586 | 0.1336 |
| RSD WITH SITE 2 |  | NA | 0.1551 | 0.1284 |
| RSD WITH SITE 3 |  |  | NA | 0.1146 |
| RSD WITH SITE 4 |  |  |  | NA |
| Equivalent Area | 0.14 | 0.161 | 0.223 | 0.263 |

**Table 3.** Quality of the mean frequency as a classifier.

|  | Site 1 | Site 2 | Site 3 | Site 4 |
|---|---|---|---|---|
| Max | 0.2000 | 0.2590 | 0.1842 | 0.2368 |
| Width @ Max/e | 0.2152 | 0.2124 | 0.4534 | 0.5068 |
| MEAN | 4.4936 | 4.4840 | 4.3766 | 4.4114 |
| MODE | 4.5176 | 4.4454 | 4.4710 | 4.3955 |
| RSD WITH SITE 1 | NA | 0.0130 | 0.0418 | 0.0704 |
| RSD WITH SITE 2 |  | NA | 0.0366 | 0.0633 |
| RSD WITH SITE 3 |  |  | NA | 0.0376 |
| RSD WITH SITE 4 |  |  |  | NA |
| Equivalent Area | 0.043 | 0.0546 | 0.081 | 0.117 |

**Table 4.** Quality of the x-x spectrum inertia as a classifier.

|  | Site 1 | Site 2 | Site 3 | Site 4 |
|---|---|---|---|---|
| Max | 0.3282 | 0.3026 | 0.2842 | 0.4684 |
| Width @ Max/e | 0.0625 | 0.0680 | 0.1582 | 0.1027 |
| MEAN | 0.0916 | 0.1022 | 0.1732 | 0.1248 |
| MODE | 0.0696 | 0.0769 | 0.1335 | 0.0677 |
| RSD WITH SITE 1 | NA | 0.0052 | 0.0447 | 0.0330 |
| RSD WITH SITE 2 |  | NA | 0.0404 | 0.0286 |
| RSD WITH SITE 3 |  |  | NA | 0.0230 |
| RSD WITH SITE 4 |  |  |  | NA |
| Equivalent Area | 0.02 | 0.020 | 0.045 | 0.0468 |

**Table 5.** Quality of the y-y spectrum inertia as a classifier.

|  | SITE 1 | SITE 2 | SITE 3 | SITE 4 |
|---|---|---|---|---|
| Max | 0.2077 | 0.2744 | 0.2105 | 0.2526 |
| Width @ Max/e | 1.9184 | 1.8314 | 3.5602 | 4.1214 |
| MEAN | 20.7688 | 20.7224 | 19.6960 | 20.1495 |

**Table 5.** *Cont.*

|  | **SITE 1** | **SITE 2** | **SITE 3** | **SITE 4** |
|---|---|---|---|---|
| MODE | 20.7680 | 20.5010 | 20.6450 | 20.1650 |
| RSD WITH SITE 1 | NA | 0.1275 | 0.3807 | 0.6215 |
| RSD WITH SITE 2 |  | NA | 0.3631 | 0.5509 |
| RSD WITH SITE 3 |  |  | NA | 0.3847 |
| RSD WITH SITE 3 |  |  |  | NA |
| Equivalent Area | 0.403 | 0.494 | 0.751 | 1.02 |

**Table 6.** Quality of the number of crossings at 1/12 max power as a classifier.

|  | **Site 1** | **Site 2** | **Site 3** | **Site 4** |
|---|---|---|---|---|
| Max | 0.3077 | 0.3872 | 0.3158 | 0.3526 |
| Width @ Max/e | 7.5381 | 5.6767 | 8.1353 | 7.0281 |
| MEAN | 6.3128 | 5.6410 | 7.9894 | 7.7368 |
| MODE | 6 | 4 | 8 | 8 |
| RSD WITH SITE 1 | NA | 0.6932 | 0.9704 | 0.8906 |
| RSD WITH SITE 2 |  | NA | 1.5542 | 1.4951 |
| RSD WITH SITE 3 |  |  | NA | 0.2297 |
| RSD WITH SITE 4 |  |  |  | NA |
| Equivalent Area | 2.34 | 2.22 | 2.6 | 2.46 |

**Table 7.** Quality of the number of crossings at 1/16 max power as a classifier.

|  | **Site 1** | **Site 2** | **Site 3** | **Site 4** |
|---|---|---|---|---|
| Max | 0.3205 | 0.4513 | 0.3158 | 0.3000 |
| Width @ Max/e | 7.2514 | 5.0034 | 7.2502 | 7.4181 |
| MEAN | 5.7436 | 5.1538 | 8.0421 | 7.4526 |
| MODE | 6 | 4 | 8 | 8 |
| RSD WITH SITE 1 | NA | 0.6007 | 1.3999 | 1.0657 |
| RSD WITH SITE 2 |  | NA | 1.8747 | 1.5903 |
| RSD WITH SITE 3 |  |  | NA | 0.3928 |
| RSD WITH SITE 4 |  |  |  | NA |
| Equivalent Area | 2.32 | 2.25 | 2.32 | 2.22 |

The analysis of Figures 5 and 6 and Tables 2–7 portrays the following on the different distributions:

### 5.1. Maximal Normalized Power Distribution (Figure 5a and Table 2)

For this parameter, the mixed RSD estimate—(between sand and sandstone sites) showed a factor of 10 that is more relative to the two sand-sited RSD. In addition, it could be noted that only sandstone shows a maximal normalized power greater than 1.8. The maximal value, mean, and mode of the distribution for the two bottom types are not significantly different. The width of the graph and its by-product—"equivalent area" is different by 30% or more.

### 5.2. Mean Frequency Distribution (Figure 5b and Table 3)

The mixed RSD estimate—(between sand and sandstone sites) showed a factor of three to six relative to the two sand sites' RSD. This is less significant than the results of the maximal power distribution (factor of 10). In addition, it could be noted that only sandstone shows a mean frequency of less than 4.2 kHz or above 4.7 kHz. The maximal value, mean, and mode of the distribution for the two types of the bottom are not significantly different. The width of the graph and its by-product—"distribution area" differ by a factor of more than two.

### 5.3. The x-x Spectrum Inertia (Figure 5c and Table 4)

The x-x inertia distribution is correlated to the maximal normalized power distribution and hence the same effect is evident in the different measures of separation. Note that similar relations characterize the RSD values (including a factor of 10 between the sand-sand RSD and all the other RSD's). In this sense, the x-x spectrum does not contribute significantly to the insights provided by the maximal power distribution.

### 5.4. The y-y Spectrum Inertia (Figure 5d and Table 5)

In the same manner, as the relation between the max power and the x-x inertial distributions, y-y spectrum inertia distribution is like the mean frequency distribution and does not bear any other insights.

### 5.5. The Number of Crossings at 1/12 Max Power (Figure 6a and Table 6)

The RSD estimate for the distribution of this parameter, contrary to all the above noted potential classifiers distributions, has the lowest value for the rock-rock case. The maximal value, mean, and mode for the two types of the bottom do not differ significantly, while the width of the graph and its by-product do not differ significantly.

### 5.6. The Number of Crossings at 1/16 Max Power (Figure 6b and Table 7)

This distribution is like that of 1/12 crossings with a slightly higher difference between the mixed RSD's (rock-sand combinations) and the uniform RSD's (rock-rock and sand-sand).

### 5.7. The Probability of Sand and Sandstone Type of Soil Based on a Measured Value of a Classifier

The above-mentioned results show the probability for a certain classifier to receive a certain value for sand and sandstone and help us to assess the potential of different classifiers. However, the classification issue is the inverse problem meaning, for a given value of a classifier what is the probability for sand and what is the probability for sandstone. This is presented in Figure 7 for the Number of Crossings at 1/16 of the Maximal Power classifier.

Figure 7 shows that when the number of crossings is 8, 10 the probability for sandstone detection is 80% and 95%, respectively. When the number of crossings is 12 or above, the probability for sandstone detection is 100%. However, when the number of crossings is 2 or 4, the probability of sand detection is 80%. Note that when the number of crossings is six there is an approximately equal probability for sand or sandstone and hence an additional classifier is necessary to distinguish between both sediments.

The probability of six crossings at the level of 1/16 of the maximal power is 26% (i.e., 26% of the spectra of all the traces, both in sand and sandstone had 6 crossings at the 1/16 of maximal power level). Hence, at 74% of the cases, it can be determined with over 80% certainty whether the bottom consists of sand or sandstone according to this parameter.

The probability for sand or sandstone for a given mean frequency value is presented in Figure 8.

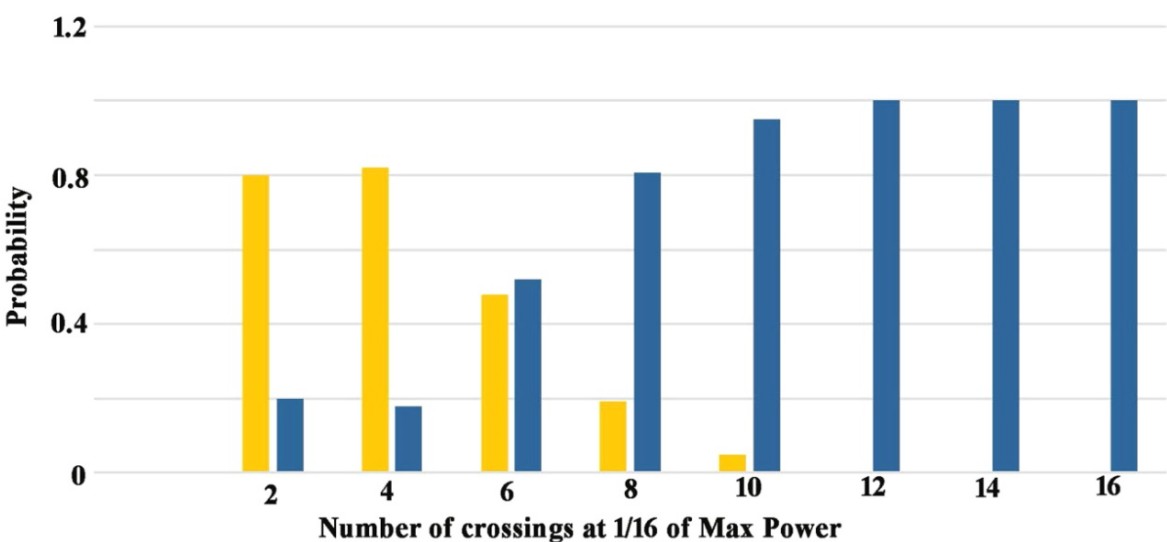

**Figure 7.** Probability for sand (yellow) or sandstone (blue) for a given number of crossings at 1/16 of the maximal normalized power.

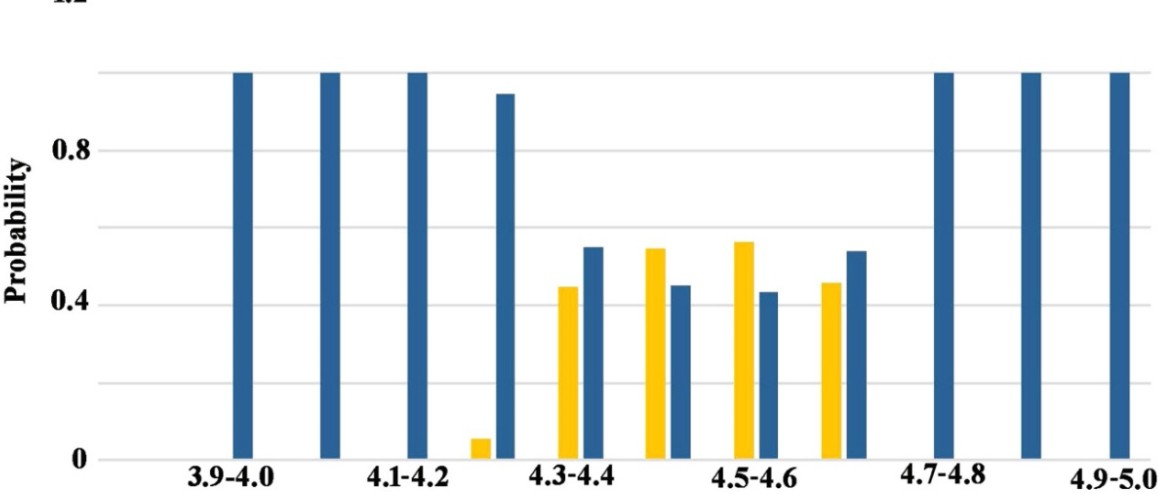

**Figure 8.** Probability for sand (yellow) or sandstone (blue) for a given range of mean frequency.

Examining Figure 8 shows that when the mean frequency is below 4.2 or above 4.7 kHz the probability for sandstone is 100%. When it is in the range 4.2–4.3 kHz the probability for sandstone is 95%. An analysis of the data shows that the probability for the sandstone reflector to have a mean frequency in those ranges is up to 13% (i.e., 13% of the spectra of all the traces, both in sand and sandstone had a mean frequency outside the range of 4.3–4.7 kHz). Only in these cases can this parameter be used as the primary classifier.

The probability for sand or sandstone for a given maximal normalized power value is presented in Figure 9. Figure 9 shows that when the maximal normalized power is above 1.9, the probability for the sandstone sea bottom is over 85% and when the maximal normalized power is above 2, the probability for the sandstone sediment is 100%. However, the probability for a reflector to have maximal normalized power above 1.9 is up to 5%. Only in these cases can this parameter be used as the primary classifier.

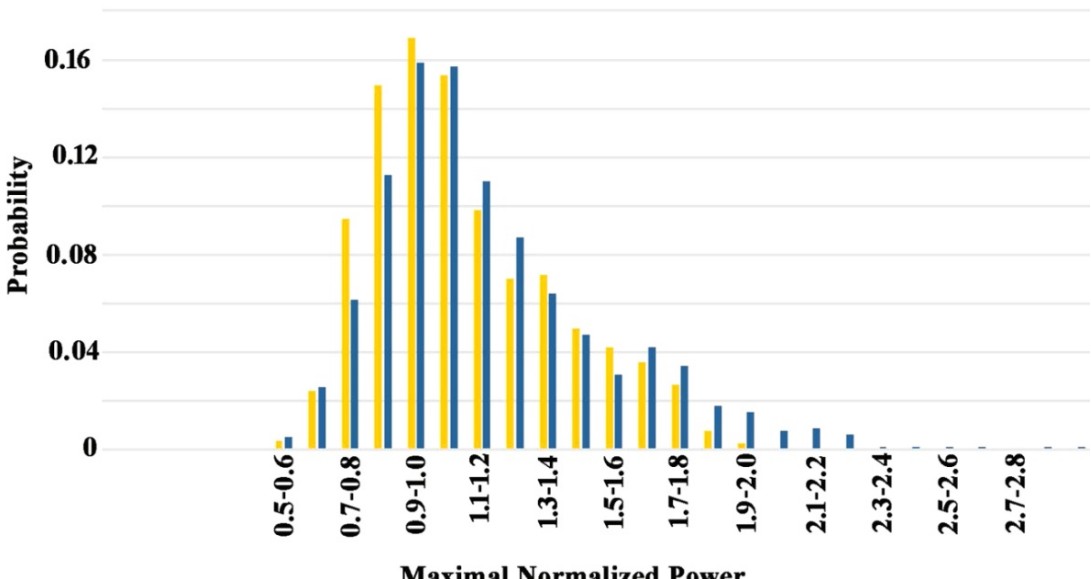

**Figure 9.** Probability for sand (yellow) or sandstone (blue) for a given range of maximal normalized power.

## 6. Discussion

From the results presented above it follows that the two sea bottom types can be characterized quantitatively via estimates (classifiers) based on acoustic measurements. Differences in the maximal normalized power (or x-x inertia), the mean frequency (or y-y spectrum inertia), and the number of crossings at 1/16 reflect differences in the bottom physical properties and their morphology.

The difference in reflected pulse inevitably follows from the combination of two phenomena. First, the acoustic impedance contrast at the water-bottom interface is much more significant in the case of sandstone compared to the case of sand. This directly affects the maximal normalized power distribution where for sandstone the distribution is wider and extends to larger values than the distribution for sand, as is clear in Figure 5a (and in a comparable manner Figure 5c). The second phenomenon relates to the geometrical features of the reflectors. The sand is relatively uniform with geometrical features (grain size, ripples, etc.) which are usually small relative to the wavelengths of the acoustic signals. This results in a more uniform spectrum, as is evident from Figure 4a. This also results in the relatively narrow band of mean frequencies for the sand, as is evident from Figure 5b (and in a similar manner Figure 5c). On the other hand, the geometrical features (discontinuities, voids, and mainly relief and stratification) of the sandstone vary in their typical length distribution (even in the small distances within the acoustic signals' footprint as well as the movement of the vessel around its mooring point). These geometrical features are of typical lengths such as a broad range of wavelengths within the incident signals bandwidth. This is the main cause for the non-uniformity (i.e., substantial number of spikes) in the reflected spectrum from sandstone, as evident from Figure 4b. This is also the cause for the wider distribution of mean frequencies in the reflections from sandstone, as is shown in Figure 5b (and in a comparable manner in Figure 5c). The effect of differences in the typical length of the geometrical features are then enhanced by the differences in contrast to the acoustic impedance (i.e., the stiffness). A comparison of the mechanical properties of sand and sandstone confirms this conclusion. The lab measurements of mechanical properties of sand and sandstone demonstrated the following results: the uniaxial compression strength (UCS) $-9 \pm 3$ and $105 \pm 5$ kPa, the value of friction angle $-39° \pm 4$ and $\sim46° \pm 2$, the value of cohesion $-1 \pm 0.5$kPa and $40 \pm 2$ kPa, tensile strength 0 and $4 \pm 1.5$ kPa, and Poisson ratio $-0.5$ and 0.29, respectively. These values are consistent with those by Collins and Sitar [47], who presented the following values for sand and sandstone properties: the

uniaxial compression strength (UCS) of sand and sandstone is the order of 13 and 124 kPa, respectively, the value of friction angle $-\sim39°$ and $\sim46°$, respectively, the value of cohesion $\sim6$ kPa and 34 kPa, respectively, and Young's modulus $\sim23$ MPa and $\sim50$ MPa, respectively. The results imply that for a stiff reflector (sandstone), the signal is more sensitive to the bottom roughness. Note that our results are mainly consistent with those by [31], who noted the essential difference between reflection properties from fine and course uncemented sediments and from quartz bearing rocks. Such a phenomenon (discrepancy in stiffness) inevitably causes the spectra reflected from sand to be smoother than it is for sandstone. The latter one is characterized by a larger number of spikes than that of sand. It allows one to use the number of crossings of the spectra at different power levels to serve as classifiers (Figures 6 and 7). In addition, the distribution of the number of crossings reaches its peak at different numbers of crossings for sandstone and sand. This results in the number of crossings being an efficient classifier. As shown in Figure 7, for the examined reflections this classifier can give 80% to 100% certainty on the type of soil whenever the number of crossings is NOT 6. Analysis of the data shows that this is the case in 74% of soundings (i.e., only 26% of the spectra of all the traces of both sand and sandstone had 6 crossings at the 1/16 of maximal power level). The cut level to which the number of crossings refer cannot be taken arbitrarily and can be the subject for calibration and additional future study in order to gain a deeper understanding of how to choose the optimal cross-level to be used to calculate the classifier and why.

The maximal normalized power, as well as the mean frequency, are also possible classifiers in certain cases. The spectra reflected from sandstone tend to have higher peaks, usually in the vicinity of 3.5 and 4.5 kHz, while the spectra reflected from sand hold smaller peaks and are more uniform due to the relative geometrical uniformity of the sand bottom (Figure 4). This results in wider maximal normalized power and mean frequency distributions for sandstone compared to those of sand. However, the distributions are centered around similar values resulting in a reduction of cases in which these classifiers are efficient. The wider distribution of mean frequency and maximal power for sandstone relative to sand results in values for these classifiers indicate over 95% probability of sandstone, as demonstrated in Figures 8 and 9 (in case sand and sandstone are the only geologically viable options). However, the over 95% sandstone probability showing values occurs only up to 13% of the time for the mean frequency and only up to 5% for the maximal power. Thus, these can be secondary classifiers to the number of crossings classifier. A combination of classifiers can be used to estimate the probability for sandstone or sand and to support a certain classification. Lastly, it is noted that the efficiency of the proposed method to distinguish between types of soil with similar acoustic impedance, and other scales of geometric features (grain size. etc.) such as sand and silt still needs to be investigated. In particular, much higher frequencies with wavelengths close to the geometrical features need to be employed. The response of these sediments to higher frequencies and the resulting adequate classifiers are the subject of future research.

## 7. Summary and Conclusions

The present study consisted of the data acquisition over two sandy sites and two sandstone sites. The spectra of the reflected signals from the sand bottom were shown to be essentially different from the reflected spectra typical for the sandstone bottom. These favorable properties stem from the dissimilarity in the acoustic impedance and the typical length of geometric features (stratification and small-scale topography). The spectra reflected from the sand sites are much smoother than those reflected from the sandstone sites. This difference can be quantified by examining the number of crossings at different power levels, the maximal normalized reflected power, and the mean frequency. The statistical value distribution of these parameters was calculated and analyzed. To summarize, we found the following:

Sand and sandstone are characterized by different spectral acoustic features in the 2.75–6.75 kHz range.

The differences in acoustic features of sand and sandstone can be characterized by classifiers such as the number of crossings at different levels (representing the roughness of the spectrum). The number of crossings at 1/16 of the maximal normalized power classifier showed the ability to assess the probability for sand or sandstone with over 80% certainty in over 75% of the cases. Therefore, it can be used as a primary quantitative measure for actually distinguishing between sand and sandstone based on spectral data collected from sub bottom profiler surveys.

The mean frequency and maximal normalized power can serve as auxiliary and supplementary classifiers, as they can give the probability for sand or sandstone with over 95% and 85%, respectively in up to 13% and 5% of the cases, respectively.

This paper deals with four experimental data sets, because its aim is to introduce a new method for marine soil classifying. It is recognized that the study presented here is only a first but obligatory step towards the goal of defining a general method for classifying marine soils based on their acoustic fingerprints. Considering that it is straightforward to apply the current method in real time, it would be interesting to evaluate the effectiveness of the method for other subsoil and rock types. To reach the goal, more statistical data has of course to be processed using this method in order to improve the criteria suggested. Therefore, the following steps are considered for future research: additional methods of processing (e.g., the maximum likelihood method) will be studied and compared with the results obtained by the FFT method; additional classifiers, the relative weight of each classifier, and the physical mechanisms resulting in classifier value distributions will be the subject of further research; to further develop the use of the suggested classifiers and to formulate a generalized classification method, future research will include the studying the spectral parameters used in this paper over a large area with different sediment types and not in a limited number of small locations.

**Author Contributions:** Conceptualization, U.K. and V.F.; Data curation, U.K.; Formal analysis, U.K. and V.F.; Methodology, U.K.; Writing—original draft, U.K. and V.F.; Writing—review and editing, U.K. and V.F. All authors have read and agreed to the published version of the manuscript.

**Funding:** This research was funded by European Union's Horizon 2020 research and innovation programme under the Marie Sklodowska-Curie RISE project EffectFact grant agreement No 101008140 and by the Sami Shamoon College of Engineering Grants No. YR03/Y18/T2/D3/Yr2 and YR03/Y17/T1/D3/Yr1.

**Informed Consent Statement:** Not applicable.

**Data Availability Statement:** All data generated and analyzed during this study are included in the article.

**Acknowledgments:** V.F. acknowledges the support within the European Union's Horizon 2020 research and innovation programme under the Marie Sklodowska-Curie RISE project EffectFact grant agreement No 101008140. All data generated and analyzed during this study are included in the article. U.K. and V.F. would like to thank Sami Shamoon College of Engineering Grants No. YR03/Y18/T2/D3/Yr2 and YR03/Y17/T1/D3/Yr1 for the financial support that allowed a thorough study of the problem.

**Conflicts of Interest:** The authors declare that they have no conflict of interest.

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
