# Peer review of "Spectral Acoustic Fingerprints of Sand and Sandstone Sea Bottoms"

_jmse, doi:10.3390/jmse10121923_

Round 1

Reviewer 1 Report

This article is more like an experiment report. In this paper, only the relation between subsurface structure and its frequency spectra is given. It is suggested to add some other studies on acoustic properties such as reflection coefficient. And please give the signal-to-noise ratio of the experiment.

Author Response

Q1 It is suggested to add some other studies on acoustic properties such as reflection coefficient.

A1. The corresponding reference (ref. 37) was added to the amended version and the values of  reflection coefficient were added to Sect. ‎4.1. Quality Control (Line 153).

 Q2. Please give the signal-to-noise ratio of the experiment.

A2. The value of signal to noise ratio was added to Sect. ‎4.1. Quality Control (Line 148).

Reviewer 2 Report

he paper deals with the acoustic spectral response of sand and sandstone sediments at the sea bottom. An acoustic data collection recording was conducted over two sand sites and two sandstone sites. The analysis of the results shows that reflections of acoustic signals from sand and sandstone sea bottom are characterized by various spectral features in the 2.75-6.75 kHz range. The differences in acoustic response of sand and sandstone can be quantified by examining the maximal normalized reflected power, the mean frequency, and the number of crossings at different power levels. The statistical value distribution of these potential classifiers was calculated and analyzed. These classifiers, and especially the roughness of the spectrum quantified by the number of crossings parameter can give information to assess the probability for sand or sandstone based on the reflected spectra and be used for actual distinction between sand and sandstone.

English language and style are minor spell check required. The paper after minor corrections can be printed in its current form

English language and style are minor spell check required. The paper after minor corrections can be printed in its current form.

Author Response

Q1. English language and style are minor spell check required. The paper after minor corrections can be printed in its current form.

A1. The manuscript has been checked for English language and style..

Sincerely,